# Study on the effect of front retaining walls on the thermal structure and outflow temperature of reservoirs

**Xiaoqian Yang[1], Youcai Tuo[1]\*, Yanjing Yang[1], Xin Wang[2], Yun Deng[1], Haoyu Wang[1]**

**1** State Key Laboratory of Hydraulics and Mountain River Engineering, Sichuan University, Chengdu, China,
**2** Power China Guiyang Engineering Corporation Limited, Guiyang, China

\* tuoyoucai@scu.edu.cn

**Data Availability Statement:** All relevant data are within the paper and its Supporting Information files.

## Abstract

The front retaining wall (FRW) is an effective facility of selective withdrawal. Previous research has not estimated the effect of FRWs on the thermal regimes of reservoirs and outflow temperature, which are crucial to reservoir ecology. For this purpose, taking the Dongqing Reservoir as a case study, a two-dimensional hydrodynamic CE-QUAL-W2 model was configured for the typical channel-type reservoir in the southwestern Guizhou Province, to better understand the influence of FRWs on the thermal structure and outflow temperature. The simulated data from January to September 2017 showed that FRWs can change the vertical temperature distribution during the stratification period, accelerate the upper warmer water release and thus decrease the strength of thermal stratification. The stratification structure changed from a single thermocline to double thermoclines in August. An FRW resulted in an average 11.8 m increase in the thickness of the hypolimnion and a 1.2°C decrease in the thickness of the thermocline layer. An FRW increased the outflow temperature by 0.4°C and raised the withdrawal elevation by 16 m on average. The longitudinal velocity increased compared with the non-FRW condition, while the maximum velocity position moved up. In addition, FRWs can continuously obtain surface warmer water without manual operation and have low investment and good construction conditions. This study can provide an available selective withdrawal idea for reservoirs with similar hydraulic conditions.

## 1. Introduction

When a dam impoundment meets a variety of water supply needs, it is bound to have an impact on the spatiotemporal variation in the water temperature in the reservoir and downstream sections [1]. Water Temperature is an essential environmental factor of reservoirs that can affect water quality, nutrient distribution and even ecosystems [2,3]. Due to the increase in water depth, slow flow rate and longer time for surface water to receive solar radiation in reservoirs, vertical temperature stratification usually occurs in deep reservoirs [4], which has different impacts on the thermal state of the reservoir area and the structure and function of the

**Funding:** PowerChina Guiyang Survey, Design and Research Institute Co. Ltd has no funding supporting in writing this paper. Wang Xin plays a role in the investigation, software, and validation. The author(s) received no specific funding for this work.

**Competing interests:** Wang Xin, one of the authors, is a graduated master student in State Key Laboratory of Hydraulics and Mountain River Engineering, Sichuan University. His tutor was the corresponding author Mr Tuo. He plays a role in the investigation, software and validation and has nothing relevant competing interests relating to employment, consultancy, patents, products in development, or marketed products, etc. The authors declare that they have no known competing financial interests or personal relationships that could have appeared to influence the work reported in this paper.

ecosystem [5,6]. In practice, stratification in reservoirs is affected by natural processes (such as meteorological conditions) [7] and reservoir operation schemes. Moreover, stratification will also lead to the phenomenon of discharged low water temperature in summer [8], significantly changing the characteristics of water temperature in downstream rivers and harming the water quality, metabolism and reproduction of aquatic organisms and agricultural irrigation [9,10]. Thus, research on the evolution law of the stratified thermal state and discharged water temperature is the basis for optimizing the allocation of reservoir water [11].

Reservoirs with a single outlet might release water that is too warm or too cold for downstream ecosystems, depending on reservoir temperature at the intake [3]. Currently, considering the severe effect of bottom-intake for withdrawal on ecology [12,13], which was commonly employed in some north temperate countries in the 1960s, selective withdrawal systems have been widely used to meet downstream temperature and other water quality objectives [14,15], such as stoplog gates [16] and temperature-control curtains (TCCs) [17]. With the objective of water quality changing, applying a selective withdrawal facility to a certain reservoir involves many factors such as the improvement effect of outflow temperature, reservoir operation and management mode, economic investment, safety performance and so on [18]. Focusing on a reservoir that has impounded water, in regard to a deep run-off reservoir, which can redistribute the natural inflow of reservoirs in a day according to the daily load change process, the water level changes less and a facility with variable withdrawal elevation like a stoplog gate or temperature-control curtain is not necessary [19,20]. A front retaining wall (FRW) may be a more suitable choice which can meet the needs of the easy operation and low cost.

The front retaining wall (FRW) is an available fixed-elevation selective withdrawal facility composed of reinforced concrete. In terms of the flow region, a wall can be set before the water intake and supported by the arms to block bottom water from entering the water intake. Usually, the top elevation of an FRW is greater than the elevation of the water intake. In summary, the FRW is a new selective facility that has been successfully used in the Dongqing Reservoir to reduce summer outflow temperature in the high-temperature period. There have been many studies on the selective withdrawal systems and facilities [21,22]. However, little attention has been specially given to the FRW, which is proven to have an impact on the ecology downstream.

Many scholars have drawn a conclusion that selective withdrawal is the most effective means of controlling the quality of water released downstream [23]. Deeper withdrawals also facilitate heat transfer in the water column, increase the thickness of the metalimnion and weaken the stratification stability [20,24]. Through experimental study and numerical simulation, TCCs with different water-retaining proportions have been found to affect the flow field and thermal characteristic in reservoirs [25]. A top-TCC is recommended for a smaller thermal stability or a cool outflow in the summer [26]. Stop-log gates can mitigate the cold temperature effect in spring and summer [27]. Moreover, stoplog gates can influence the flow patterns including the flow regime and velocity distribution at the intake and the flow velocity at the trash rack section varies vertically in a great manner [16]. Thus, for distinct water management requirements, both relevant selective withdrawal systems and corresponding thermal properties may have a certain gap. As a new engineering measure, an FRW is mainly designed for the controlling of water released downstream. Tuo et al. illustrated the remained old dam dismantled above 240.2 m asl, standing before the new dam and acting as an FRW in Fengman Reservoir (in northeast China), can strengthen the flow motion in the surface layer and lower the outflow temperature in winter [28]. However, no studies have quantified the influence of FRWs on the thermal characteristics in reservoirs, outflow laws and hydraulic characteristics, which are crucial to the reservoir water environment.

Dongqing Reservoir is a deep and impounded reservoir in China. An FRW was initially recommended to regulate the outflow temperature, and its effect on the thermal structure is unknown. Taking Dongqing Reservoir as a case study, this paper aims to analyze the flow and temperature field of the reservoir under FRW scenarios and to obtain their potential effects on the thermal structure and outflow properties.

In this study, field observations were carried out in the Dongqing Reservoir from January to September 2017, and a two-dimensional hydrodynamic CE-QUAL-W2 model was established to simulate the vertical water temperature in front of the dam and discharged water temperature with and without an FRW. The stratification characteristics, thermal stability and hydrodynamic characteristics of the water temperature were compared and analyzed. This study can provide a new idea and strategy for the management of selective withdrawal in deep reservoirs.

## 2. Materials and methods

### 2.1 Study site and data acquisition

Dongqing Reservoir (25˚ 31'N, 105˚ 46'E) is a narrow, long and deep reservoir located in southwestern Guizhou Province, China, with a surface area of 13,500 km$^2$, a maximum dam height of 149.5 m, a maximum water depth of 120 m, an average annual runoff of approximately 12.56×10$^9$ m$^3$ and a storage capacity of 9.55×10$^8$ m$^3$ under 490 m asl. The reservoir is a typical subtropical reservoir with an average air temperature of 16.6˚C/yr. The average yearly flow in the reservoir is 398 m$^3$/s. The main reservoir is located on the Beipan River with a backwater length of 41 km, while the tributary is the Dabang River with a backwater length of approximately 26 km. A tower inlet is adopted by the power station with a floor elevation of 455 m asl. A cantilever-type reinforced concrete retaining wall is set at 13.5 m in front of the inlet, and both sides of the retaining wall are connected to the inlet side pier. The top elevation of the wall is 470 m asl with height of 15 m and thickness of 2 m (Fig 1).

To understand the thermal characteristics of this reservoir, we monitored the water temperature from January to September 2017, including vertical profile measurements (Fig 1B1 at predams), inflow water temperature (Fig 1A1 and 1A2), and outflow water temperature (Fig 1A3). A vertical thermal chain was set at section B1 (900 m upstream in front of the dam) for long-term high-frequency monitoring using a TIDBIT V2 (UTBI-001) temperature recorder (Onset, USA) at a measuring interval of 1 h. The vertical intervals of 43 water temperature sensors for the chain were 1 m (0–10 m depth), 2 m (10–50 m depth), and 4 m (50–100 m depth). The water temperatures of inflow and outflow were measured by a ZDR self-recording thermometer. The inflow observation section was settled at the end of the mainstream backwater (A1) and tributary backwater (A2), and the outflow monitoring section was in the release water of Dongqing Power Station (A3) at a measuring interval of 1 h.

### 2.2 Numerical methods and simulations

In this paper, an unsteady two-dimensional model (along the longitudinal and vertical directions) is the minimum requirement to study spatial-temporal thermal regime variations in the Dongqing Reservoir (Fig 1), which is a long-but-narrow reservoir with a longitudinal gradient in temperature that is not small enough to be neglected. Consequently, we conducted the present study by using CE-QUAL-W2, a two-dimensional laterally averaged hydrodynamic and water quality model [29]. This model is mainly applied to water bodies with longitudinal and vertical temperature gradients. It provides accurate simulations of stratification and density currents and has been widely used to study various reservoirs and river impoundments worldwide [30,31].

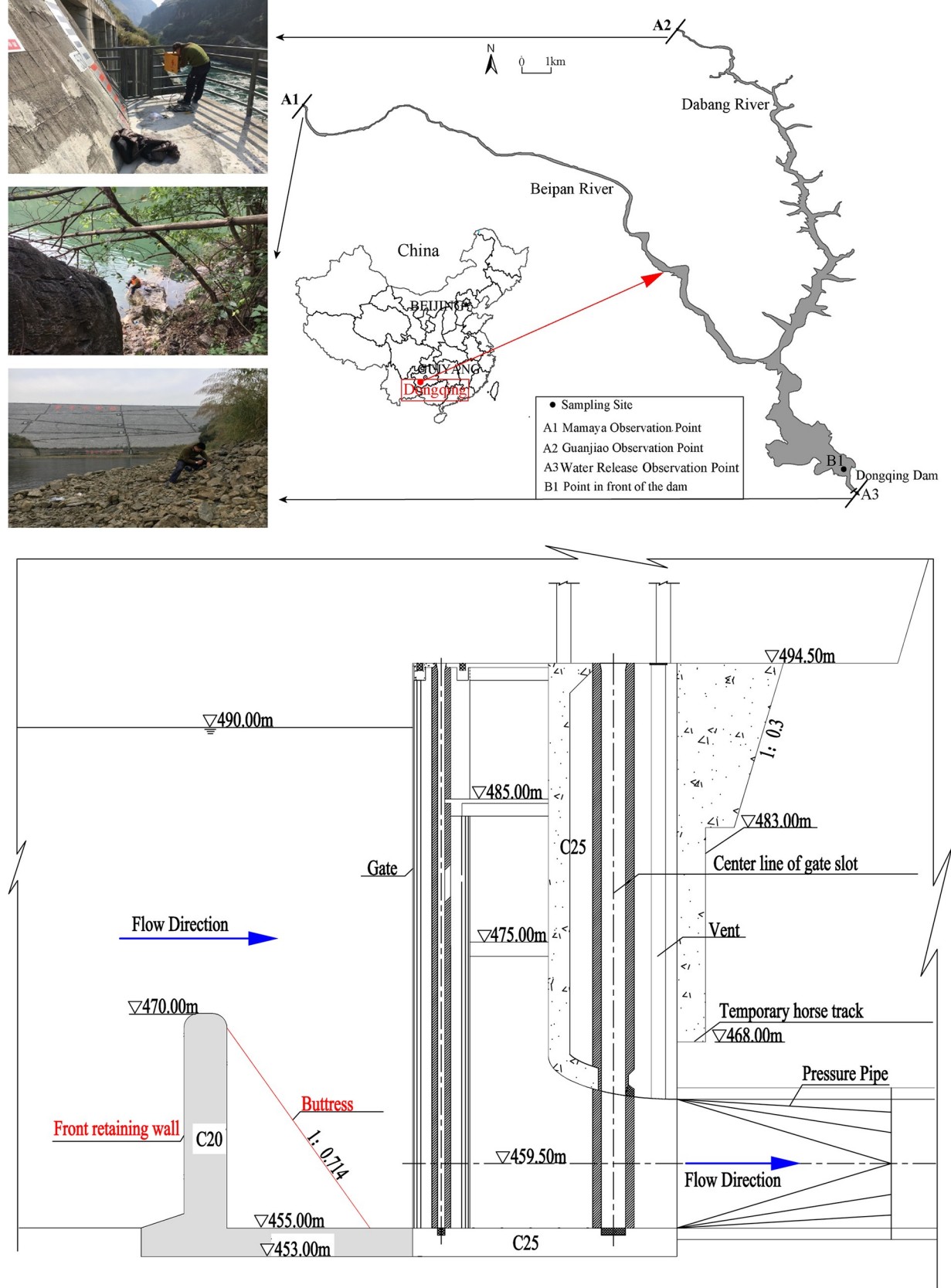

**Fig 1. Map of study locations and the structure of the water intake of Dongqing Hydropower station.** (a) Map of study locations. (b) FRW and the structure of the water intake.

We use this model to quantify thermal stratification and hydrodynamics in the Dongqing Reservoir. This model is a water quality and hydrodynamic model developed by the U.S. Army Corps of Engineers and Portland State University [29], mainly for rivers, estuaries, lakes and reservoirs in the longitudinal or vertical direction.

We constructed a longitudinal and vertical two-dimensional water temperature model for the whole reservoir area, and it ignored the variation in various variables along the width. The Dongqing Reservoir was divided into $198 \times 86$ (longitudinal × vertical) rectangular cells, with 124 cells for the main reservoir and 74 cells for the branch reservoir. The longitudinal size of the cell grid was 10–500 m, and the vertical size was 1–2 m. The input boundary (S1 Fig) included relative operation data (inflow, outflow, inflow water temperature and water level) and meteorological data (including air temperature, solar radiation, humidity, wind speed, cloud and wind direction). The temperature field at the initial moment was obtained after interpolation based on the vertical water temperature measured in the reservoir area on January 1, 2017. To explore the effect of the retaining wall on the thermal state of the reservoir area and the improvement in the low discharged water temperature, we use a numerical model to simulate the water temperature and hydrodynamics under two conditions with and without a retaining wall. The condition with the retaining wall is defined as Case 1, while the condition without the retaining wall is defined as Case 2.

## 2.3 Index of thermal stratification and statistics analysis

We selected indicators to quantify the thermal stratification of water bodies and evaluate the characteristics of thermal structure of stratification. The vertical water temperature gradient (VTG, ˚C/m; 0.12˚C/m was defined as the threshold of strong stratification here (Fig 2) [32] and the buoyancy frequency (N, 1/s) [33] was calculated based on the water temperature profiles of the vertical section in front of the dam.

$$VTG = \frac{\partial T(z)}{\partial z} \tag{1}$$

$$N = \sqrt{-\frac{g}{\rho_0}\frac{\partial \rho(z)}{\partial z}} \tag{2}$$

where T(z) is the water temperature at depth z, ˚C; $\rho(z)$ is the density at depth z, kg/m³; $\rho_0$ is the average density of the whole water column, kg/m³; g is the acceleration of gravity, m/s²; and N is an important indicator used in limnology, oceanography and human-made reservoirs. The change trend of N can be used to evaluate the stratified stability of water columns. N indicates that in a stable temperature-stratified structure, the fluid particles move in the vertical direction after being disturbed. The combined effect of gravity and buoyancy always returns these factors to an equilibrium position, which oscillates due to inertia. The oscillation frequency can be understood as the exchange rate of water.

To better evaluate the changes in the stratification stability of the water column, the SI index, which reflects the stability of the deep water body, was used to evaluate the strength of the stratification stability of the reservoir [34].

$$SI = \sum_{Z=Z_0}^{Z=Z_1}(Z - \bar{Z})\rho_Z \tag{3}$$

where Z is the depth of the water column from the surface; $Z_0$, $Z_1$ and $\bar{Z}$ are the depths of the surface water, the lower end of the water column, and the centroid of the water column,

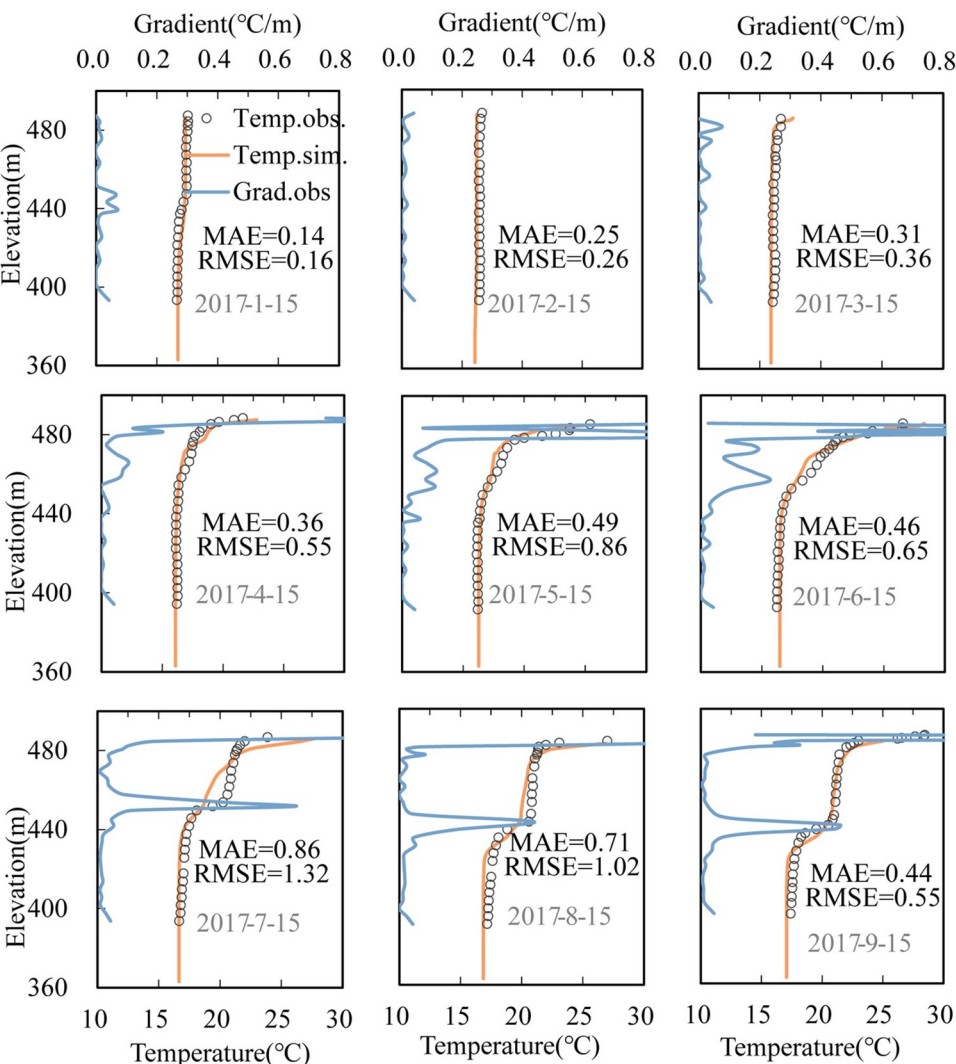

**Fig 2. Comparison of simulated and observed values of vertical temperature distribution.** The line represents the calculated values, while the dots represent the measured values.

respectively; and $\rho_z$ is the water density at depth Z. SI can be converted into energy by multiplying by the acceleration of gravity and the volume of each layer, and SI represents the ideal energy estimate of the entire water column to achieve mixing in the depth range without an increase or decrease in heat.

We used the root mean square error (RMSE) and mean absolute error (MAE) to evaluate the goodness of the fitted temperature curve, as well as the differences between the observed and simulated temperatures. A paired T-test was conducted to determine the significance levels of the differences in the (a) water temperature structure and (b) stratification stability index between two different withdrawal conditions. A value of $P < 0.01$ was defined as significant. Moreover, the correlation coefficient $R^2$ was employed to assess the goodness of fit between the measured and simulated values of discharge water temperature with the retaining wall. The closer $R^2$ is to 1, the stronger the data correlation and the higher the simulation accuracy is.

**Table 1. Calibrated parameters in the CE-QUAL-W2 model of the Dongqing reservoir.**

| Parameter | Unit | Value |
|---|---|---|
| Longitudinal eddy viscosity | m²/s | 1 |
| Longitudinal eddy diffusivity | m²/s | 1 |
| Wind sheltering coefficient | - | 1 |
| Extinction for pure water | m⁻¹ | 0.5 |
| Solar radiation absorbed in the surface layer | - | 0.65 |
| Dynamic shading coefficient | - | 1 |

## 3. Results and discussion

### 3.1 Model calibration and validation

The calibration of Dongqing Reservoir's CE-QUAL-W2 model focused on several critical coefficients that had the greatest influence on the simulated temperature profiles upstream of the dam. The critical model parameters are the longitudinal eddy viscosity coefficient, longitudinal eddy diffusion coefficient, wind shielding coefficient, extinction for pure water, solar radiation absorbed in the surface layer and dynamic shading coefficient (Table 1). The calibration and value of these parameters referred to previous research [32,35,36].

We compared the calculated results and measured data for upstream of the dam (Fig 2) and outflow temperature (Fig 3) based on hydrological and meteorological data from January to September 15, 2017, which were obtained from the monitoring of the upstream and downstream power station of the Dongqing Reservoir and the meteorological website (http://www.cma.gov.cn/2011qxfw/2011qsjgx/), respectively. The temperature variation trends of the epilimnion and metalimnion were consistent, and the temperature of the hypolimnion was generally well matched. The average MAE and RMSE of the vertical water temperature in front of the dam from January to September were 0.46°C and 0.62°C, respectively. This was mainly because the meteorological data originated from the Zhenfeng Meteorological Station. The station was approximately 40 km away from the dam site whose wind speed and direction deviated from the wind conditions in the open area of the reservoir area. Although we calibrated the wind shielding coefficient to be 1.0, there may still be a certain deviation in wind direction and speed. However, these errors are within a reasonable range, with relative errors of 4.4% and 3.5% in July and August, respectively.

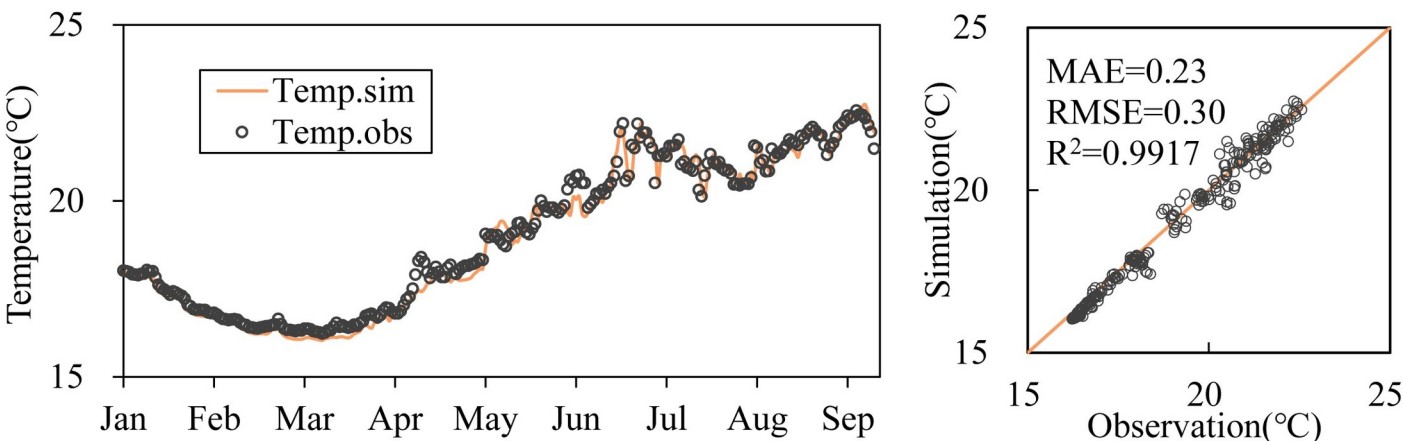

**Fig 3. Comparison of simulated and observed values of outflow temperature.** The line represents the calculated values, while the dots represent the measured values.

The scatter plot of the measured and calculated outflow temperatures (Fig 3) showed that the degree of dispersion of the error was small with an MAE of 0.23°C and RMSE of 0.3°C. The determination coefficient between the measured value and calculated value was 0.9917. The results showed that the model can accurately simulate the effects of buoyancy flow and atmospheric heat exchange on the thermal structure of water temperature.

## 3.2 Effect of a retaining wall on thermal stratification

**3.2.1 Temperature profiles in different cases.** Fig 4 compares the vertical water temperature distribution in front of the dam with or without an FRW from January to September 2017. The results showed consistent seasonal variation characteristics under both conditions. Stratification began during the warming period (April). Until the high-temperature period (August), the maximum temperature difference between the surface layer (epilimnion) and bottom layer (hypolimnion) increased to 11°C. A double thermocline structure existed simultaneously because of a larger inflow discharge, whose temperature was lower than that of the epilimnion and dived along the reservoir into the layer with the same density, forming a horizontal intrusion flow. The initial momentum of inflow during the submersion process was large, leading to severe turbulent diffusion and a strong mixing effect, thus forming an isothermal layer with a certain thickness and the middle thermocline.

The stratification structure under the two cases differed greatly for different withdrawal modes, mainly in the stratification period (Fig 4, paired T-test: P < 0.01, April to September). With the development of stratification, the water temperature isoline gradually extended to the bottom owing to the increasing air temperature and heat storage (Fig 6A and 6D), and in case 2, it extended much earlier to the bottom. The average vertical temperature of the whole profile varied from 16.1°C to 28.3°C in case 1, while that of case 2 ranged from 16.2°C to 34°C. This is because deeper low-temperature water was discharged leading to more heat storage capacity when there was no FRW. The thickness of the thermocline in case 1 was 8 m-16 m smaller than that in case 2, and the thickness of the hypolimnion exhibited the opposite trend (8 m-14.5 m). For example, during the double thermocline period in August, the depth of the middle isothermal layer in case 1 was 12 m higher than that in case 2 owing to the lower water temperature in the epilimnion and thermocline resulting in a smaller diving depth of the inflow. The thickness of the hypolimnion was 58 m in case 1 and 50 m in case 2.

**3.2.2 Thermal stratification strength characteristics.** Selective withdrawal can change the thermal stratification intensity of reservoirs [37]. We can see from Fig 4 that the variation rule in N was consistent with the VTG, and the peak value of N is the location of the thermocline (Fig 4). ΣN represents the overall buoyant flow strength of the water column. The larger ΣN was, the greater the density stratification strength. Thus, the density stratification in case 2 was more intense than that in case 1. In August, ΣN was 0.724 and 0.921 in case 1 and 2, respectively.

Fig 5 present the temperature contour, VTG and N variations upstream of Dongqing Dam for the two cases over time. In April, the stratified flow developed initially with changes in meteorological conditions and inflow conditions. From May to June, stratification strengthened. The water temperature at a depth of 10 m increased rapidly, and the density decreased. Influenced by the buoyancy jacking effect, N varied from 0.02 $s^{-1}$ to 0.08 $s^{-1}$. From July to August, the inflow of the reservoir increased (798 $m^3$/s), and the solar radiation increased (186 W/$m^2$ on average per month). In case 1, the main flow layer was located near 470 m, and N increased from 0.02 $s^{-1}$ to 0.067 $s^{-1}$. The inflow mixed quickly within a range of 455 m to 465.5 m, forming an isothermal layer. Moreover, the epilimnion appeared with an N of 0.028 $s^{-1}$. The temperature of the hypolimnion remained stable and was no longer affected by the

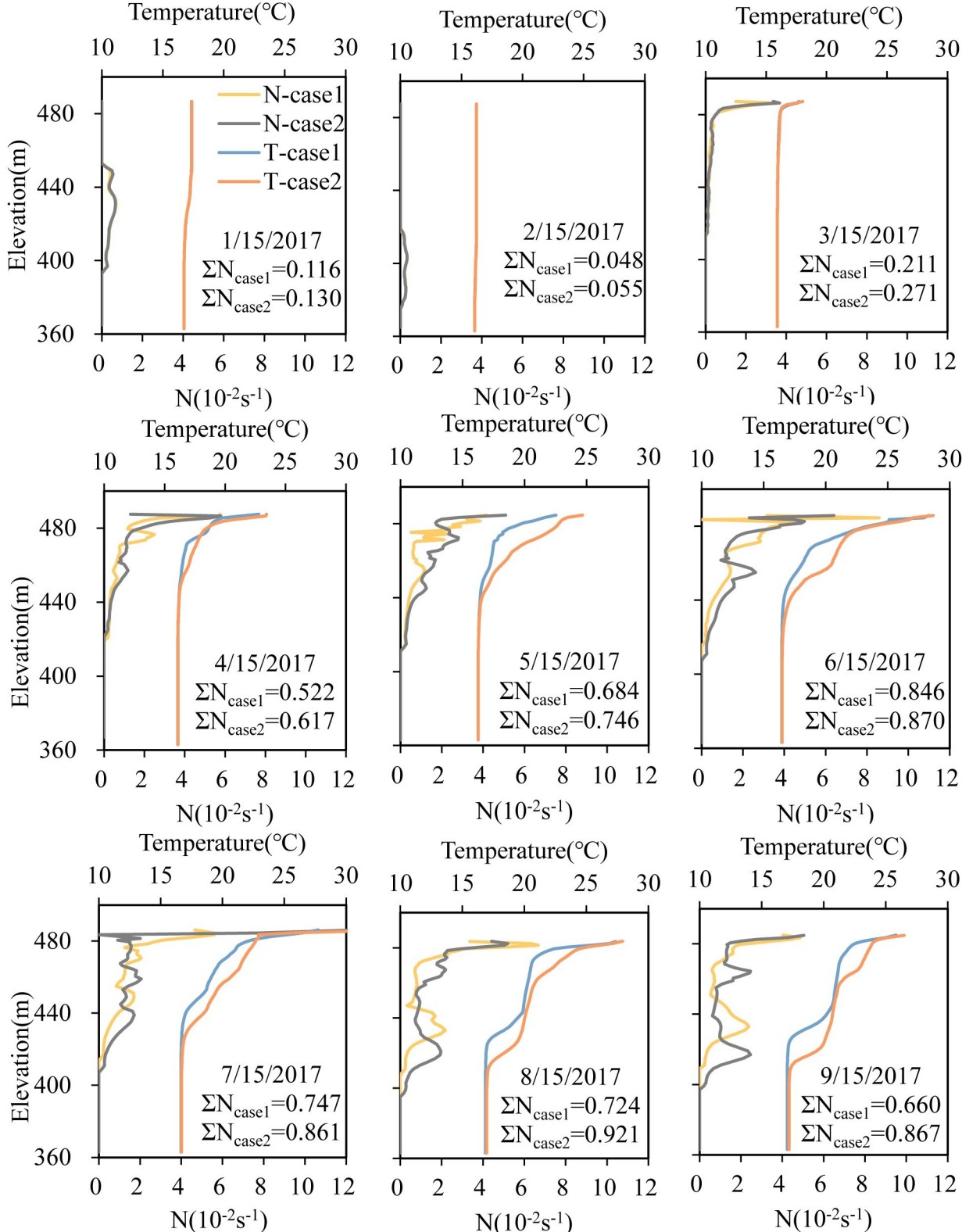

**Fig 4. Temperature profiles at 1–2 m intervals simulated based on different cases and the corresponding buoyancy frequency (N) for the entire water column during 2017.** $\Sigma N$ indicates the total stratification intensity.

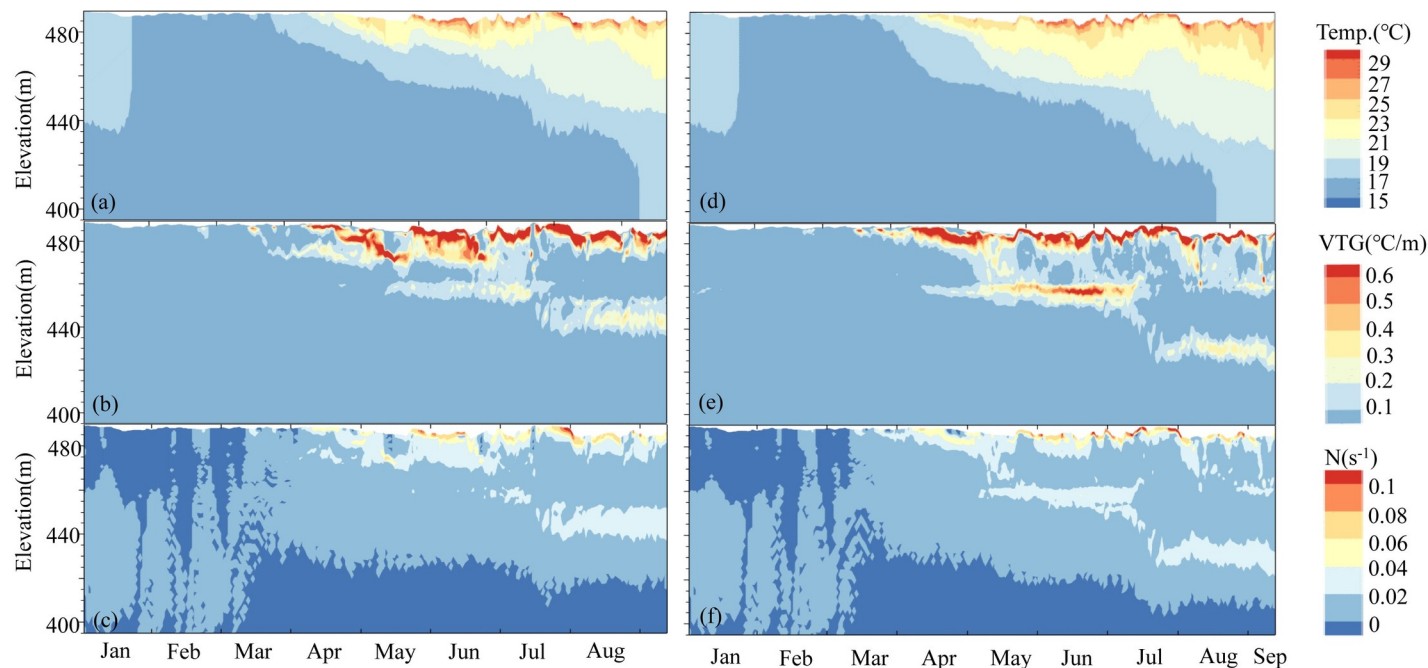

**Fig 5.** From January 1 to September 15, 2017, the contour profile of the cross section (B1) 0.9 km upstream of Dongqing Dam for two conditions (a-c for case 1 and d-f for case 2) and (a, d) water temperature contour, (b, e) vertical temperature gradient and (c, f) buoyancy frequency.

turbulence caused by the FRW. The water column presented a density stratified flow state between layers. Comparatively, in case 2, the mean plug flow was located near 458 m with N of 0.02 s$^{-1}$. The depth of the middle thermocline decreased significantly, and the surface water was supported by buoyancy, while the hypolimnion was compressed to below 420 m asl.

Fig 6 compares the daily variation in SI in two cases (kg/m$^2$). Their annual variation trends of reservoir SI were similar, with averages of 402 kg/m$^2$ (case 1) and 516 kg/m$^2$ (case 2). As the temperature rose, the stratification intensity continued to increase. Stratification of the water column gradually strengthened from weak stratification to stable stratification. The SI value

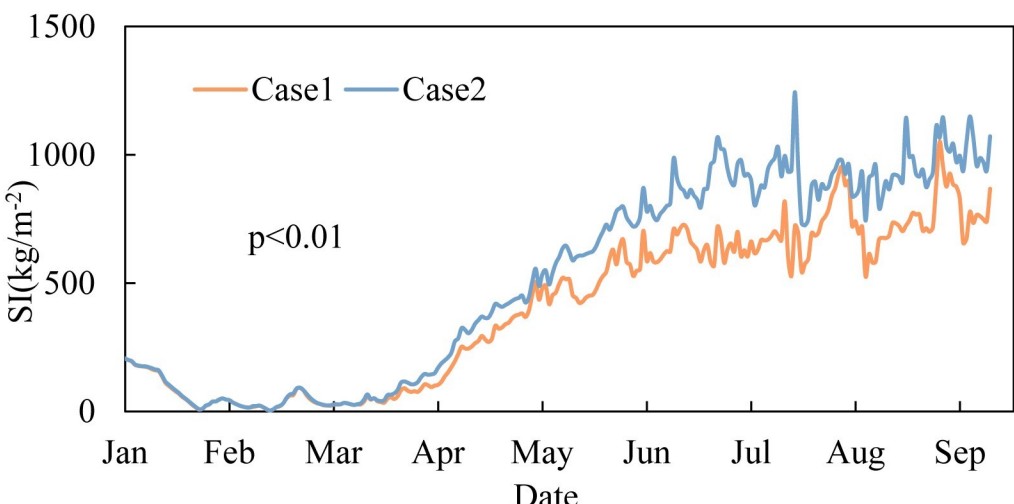

**Fig 6. Variation in reservoir SI from January to September 2017 in different cases.**

was significantly distinct in the two cases (p<0.01). In case 1, the temperature of the epilimnion was lower and the maximum temperature difference between the epilimnion and hypolimnion decreased. The ability to resist heat convection, surface wind stress and density convection caused by low-temperature inflow was relatively weaker, so the water column stability was weaker. With the arrival of high temperatures in the flood season, the replacement of water bodies was accelerated, while stratification was not weakened. The SI difference between the two cases reached a maximum in July, with an average difference of 522 kg/m$^2$. This was because of the sudden increase in flow in July and the weaker stratification stability in case 1, resulting in more intense mixing than in case 2.

### 3.3 Influence of the retaining wall on outflow temperature

**3.3.1 Variations in outflow temperature.**   In subtropical reservoirs in China, many warm-water fish (such as four major Chinese carps) are quite sensitive to outflow temperature in summer and autumn [38], and the discharged low-temperature water affects the spawning, reproduction and migration of fish. The discharged water temperature is related to the vertical temperature distribution of predams, the flow velocity near the water intake, and the flow [39]. A comparison of discharge water temperature under different working conditions is shown in Fig 7. The FRW has an obvious influence on the outflow temperature (p<0.01). Compared with the natural water temperature of the dam site, the discharge water temperature was lower in the two cases from March to September, while the degree of low-temperature water was weakened in case 1. Due to the effect of the FRW, the discharged water temperature from March to June increased by 0.1˚C~1.0˚C. The maximum difference appeared in May (1˚C). Due to the effect of meteorological conditions, the water temperature near the top elevation of the FRW (470 m asl) increased rapidly while the mixing caused by thermal convection did not move down to the vicinity of 455 m asl. The difference in water temperature near these two elevations reached the largest value. August was the double thermocline period, and the middle isothermal layer where the interlayer flow was located was near 455 m asl. There was no significant temperature difference between the middle isothermal layer in case 2 and the upper thermocline in case 1, so the outflow low-temperature water only improved by 0.3˚C. In general, the FRW mitigated the influence of low-temperature water in spring and summer at Dongqing Hydropower Station.

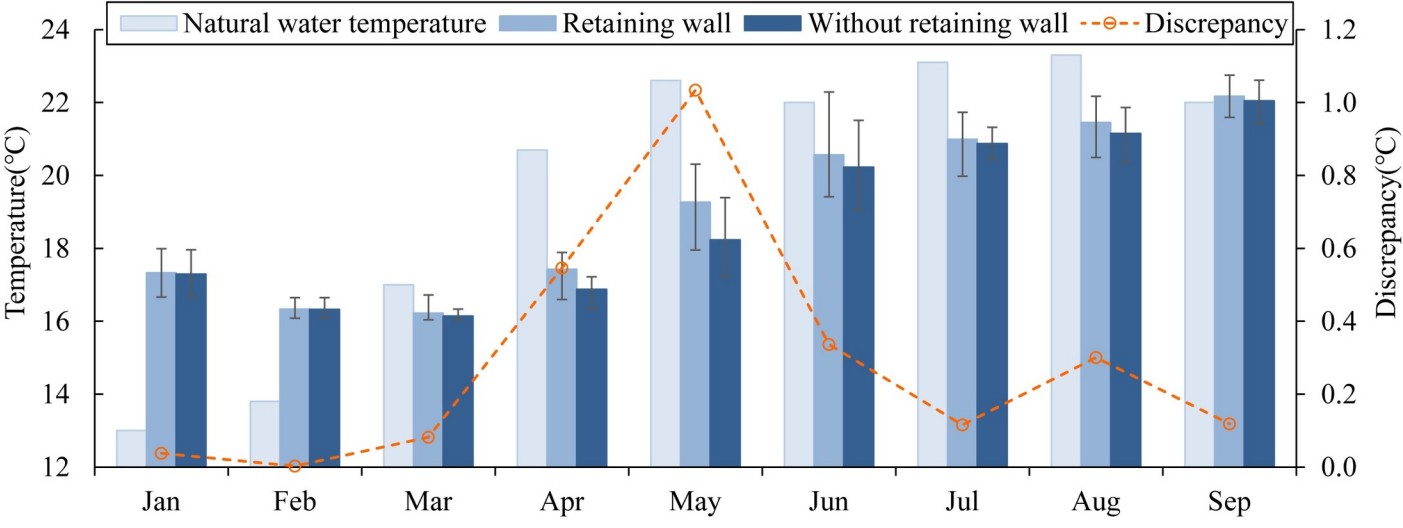

**Fig 7. Released water temperatures in different cases.**

**3.3.2 Flow velocity distribution.** The flow velocity distribution is an essential indicator of the hydrodynamic characteristics, and it is usually characterized by the maximum flow velocity and its position. In this paper, to examine the velocity distribution to analyze the withdrawal properties, we selected April in the heating period, July in the high-temperature period and September in the cooling period to explore the flow velocity changes in different periods caused by the FRW (Fig 8).

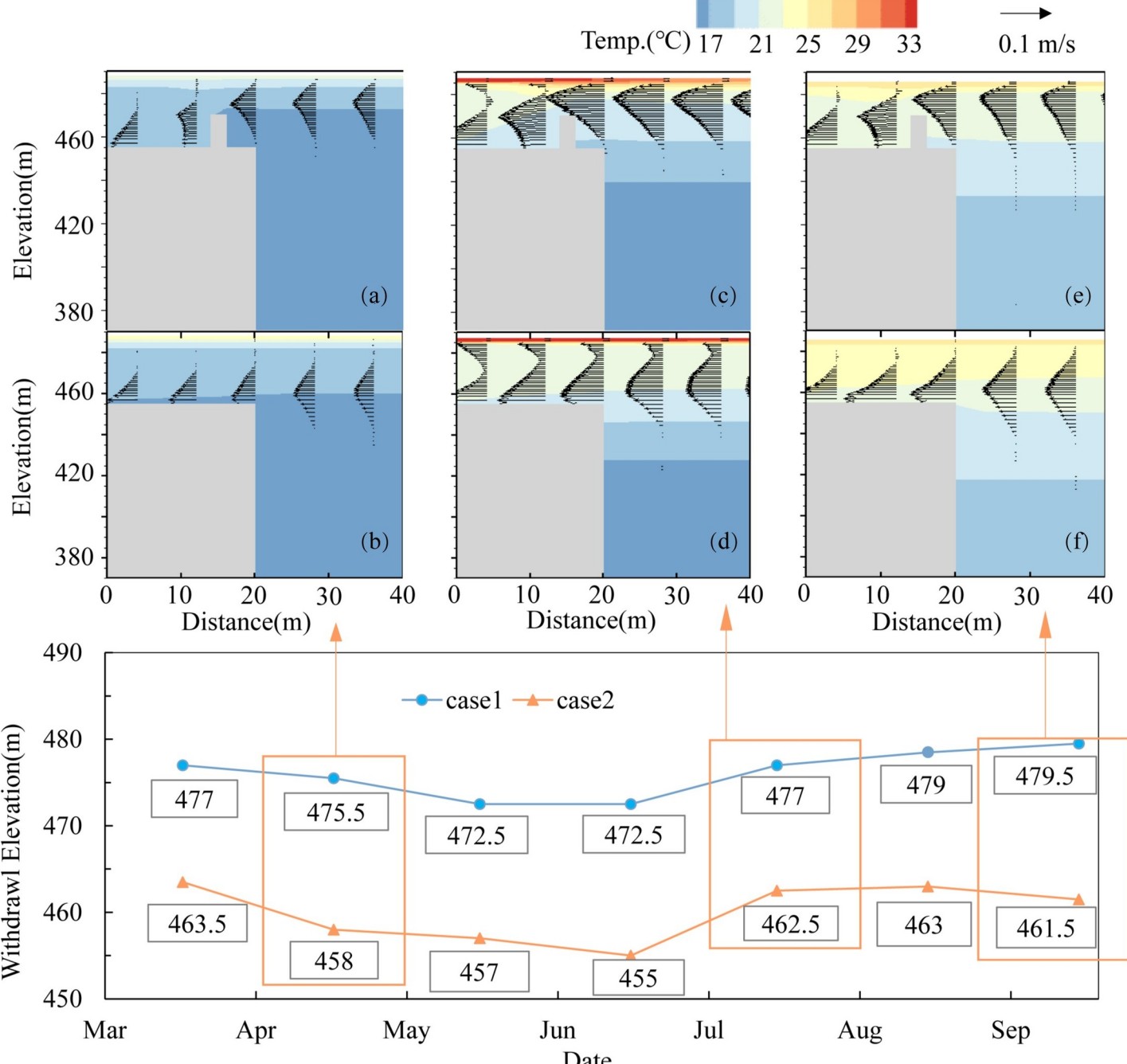

**Fig 8.** Elevations of the withdrawal layer in different months and different working conditions and the local flow field distribution in front of the dam in April, July and September: a, c and e show the flow field with the retaining walls, and b, d and f show the flow field without retaining walls.

In Fig 8A~8F, the results showed that the FRW raised the withdrawal elevation by 16m on average. The withdrawal elevation ranged from 472.5 m to 479 m asl with the FRW and ranged from 455~462 m asl with no-FRW. The FRW hindered access to water bodies below 470 m asl, allowing water bodies above 470 m asl to flow into the water intake. In case 2, the water flowed smoothly into the water intake while the flow pattern near the top of the FRW was more complex in case 1. The position of the water withdrawal layer was elevated to a warmer layer with a higher temperature, leading to more warm water release and more heat loss, which further changed the vertical water temperature structure in front of the dam (Fig 5) and lowered the upper water temperature. For example, in July, when the flow reached its peak and the increasing amplitude of the withdrawal elevation was at the minimum (14.5 m), the water temperature of the 477 m withdrawal layer in case 1 was 21.1˚C, which was lower (1.3˚C) than that in case 2. Therefore, the discharge water temperature only increased by 0.1˚C.

The layer with the maximum flow rate is recognized as the mean plug layer. In case 2, the quantity of water taken from the upper and lower layers of the mean plug layer was relatively even while the opposite was observed in case 1. In case 1, more water was taken from the upper layer, and the mean plug layer was compressed, which increased the maximum velocity. For example, in April, the center elevation of the mean plug layer with an FRW was 15 m higher than that without an FRW. The maximum velocities in case 1 and case 2 were 0.063 m/s and 0.055 m/s, and the thicknesses of the mean plug layer were 30 m and 35 m, respectively; in July, the flow reached the maximum, the velocities also increased to 0.146 m/s and 0.101 m/s, and the thicknesses of the withdrawal layer were 33 m and 39.5 m for cases 1 and 2, respectively. The release of warm water from the upper layer intensified as the flow rate increased.

**3.3.3 Equivalent withdrawal elevation.** The discharged water temperature is closely related to the vertical water temperature profile in front of the dam and the water withdrawal elevation [40,41]. Generally, when the stratification in front of the dam is obvious and the elevation of water intake is high, the discharged water temperature is high. To more intuitively express and analyze the position difference in the withdrawal layer, the equivalent withdrawal elevation is defined in this study as the elevation of the withdrawal layer corresponding to the same water temperature in front of the dam. The results show that the corresponding distributions of the water temperature and temperature gradient at equivalent elevations in the two cases are shown in Fig 9. From the distribution of the equivalent elevation, we can see the

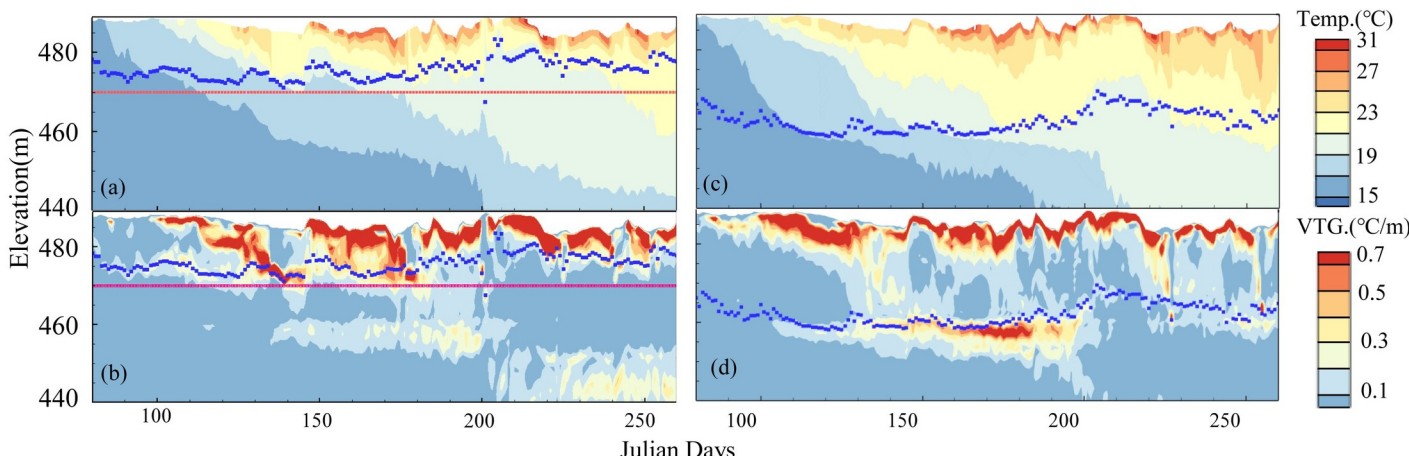

**Fig 9. Position of the equivalent elevation point in the water level-water temperature and water level-temperature gradient distribution diagram.** a and b are the water temperature and VTG distribution diagram with a retaining wall, respectively, while c and d are the water temperature and VTG distribution diagram without a retaining wall, respectively. The red dotted line represents the position of the top elevation of the retaining wall, and the blue scattered points represent the equivalent elevation point.

location of the withdrawal layer. The red line in the figure is the top elevation of the retaining wall while the blue spots represent the equivalent elevation points.

When there is an FRW, the withdrawal layer is above the retaining wall (near 470 m), while the withdrawal layer is below the intake (near 455 m) without an FRW. In July, since the inflow and outflow were large, the turbulence near the intake was intense, and the position of the withdrawal layer increased accordingly. In the two cases, the equivalent elevation points were all located at the bottom of the thermocline because the density gradient of the layer adjacent to the thermocline boundary was larger and the buoyancy effect was stronger than that of other layers (Fig 9C and 9D). The vertical mixing caused by density convection and turbulent diffusion was suppressed, and the downward transfer of surface energy was hindered at the boundary of the thermocline. The discharged water temperature was closely related to the water withdrawal elevation, so it was an effective means to improve the outflow temperature by first increasing the water withdrawal elevation. Second, the mixing of the upper warm water layer is strengthened to reduce the temperature difference in the thermocline layer, thus weakening the inhibitory effect of buoyancy.

## 3.4 Engineering applications and prospects for this study

As a new type of selective withdrawal facility, what makes front retaining walls mainly differ from other selective withdrawal facilities is that the fixing the FRW elevation makes it unnecessary for additional manual operation [42], which can provide a new idea for the adaptive water temperature management of run-off reservoirs with a relatively low amplitude. Generally, the water level of reservoirs with different operations varies greatly. The stoplog gate and temperature-control curtain can be lifted or lowered to adjust the water temperature with the water level changing [35,43–45]. However, the water level in run-off reservoirs varies less. No frequent manual operation is required to obtain the same outflow temperature. Therefore, the FRW is more suitable for this kind of reservoir.

Second, this study shows that the improvement effect of the outflow temperature with the FRW is good. For example, the water level of the Dongqing reservoir ranges from 484m to 490m. The lowest elevation of the mean plug flow is 472.5 m (June, Fig 8) near the upper thermocline, causing that the upper high-temperature water body can be continuously obtained. The maximum improvement of discharged water temperature can reach 1°C, which has a good improvement effect in engineering management. From the aspect of engineering, FRWs are achieved with a small investment and have simple operation conditions and good construction conditions, which can provide a reference for the selection of stratified water intake measures.

At present, when various selective withdrawal facilities are applied, such as stoplog gates and temperature-control curtains, it is necessary to initially determine the period for discharged low-temperature water initially. The withdrawal elevation determines the elevation conversion of the water intake. As the water level is below the submerged depth of the water intake, the intake is switched from high to low, which usually results in a sudden decrease in the outflow temperature [46]. The water temperature after lowering is lower than that without selective withdrawal facilities. Such a low temperature often causes much more severe impacts on downstream ecosystems and agricultural irrigation. Due to the fixed elevation of the front retaining wall, we can continuously obtain upper layer warm water, which will not result in sudden low-temperature water. The front retaining wall is set at a certain distance in front of the dam, so compared with the fixed intake, the head loss is relatively small, and the influence on power generation efficiency is small.

## 4. Conclusions

FRWs are a new and effective method to regulate the outflow temperature. To strengthen research on FRWs, this paper numerically investigated the effect of FRWs on the thermal structure in the deep Dongqing Reservoir and evaluated the potential function of FRWs in water environment management. The conclusions are described as follows:

1. The retaining wall significantly changed the water temperature distribution structure and reduced the thermal stability strength of the reservoir during stratification. The retaining wall increased the thickness of the thermocline layer to further restrain the downward transfer of heat. Simultaneously, the water temperature and the thickness of the upper warmer water layer were reduced. The stratification strength of the thermocline layer also decreased. ΣN was less than that without retaining walls, and SI was also relatively lower with a maximum difference of 522 kg/m$^2$ (July).

2. The retaining wall can increase the discharge water temperature during stratification by 0.4˚C on average and reaching the maximum (1˚C) in May. Specifically, in the period of the double thermocline (July and August), the discharge temperature only increased by 0.2˚C on average. The location of the mean plug layer ranged from 472.5 m to 479 m asl with the FRW and ranged from 455 m~462 m asl with no FRW. The equivalent withdrawal elevation points were distributed at the bottom of the thermocline. If the discharged water temperature rises, the water intake elevation can be lifted to reduce the temperature difference in the thermocline and weaken the inhibiting effect of buoyancy.

3. Retaining walls are more suitable for stratified deep run-off reservoirs with stable water levels without frequent manual operation. The upper warmer water can be continuously obtained, and the outflow water temperature will not suddenly decrease, resulting in more severe impacts on downstream ecosystems and agricultural irrigation during stratification. Moreover, front retaining walls have the advantages of low investment, high safety performance and good construction conditions, so they can be applied to the water temperature management of reservoirs and provide a reference for reservoirs with similar hydraulic conditions.

## Supporting information

**S1 Fig. Input boundary of the model.**
(TIF)

## Acknowledgments

I sincerely thank my coauthor Yanjing Yang, Youcai Tuo, Yun Deng, Xin Wang and Haoyu Wang for the help in my paper.

## Author Contributions

**Conceptualization:** Youcai Tuo, Xin Wang, Yun Deng.

**Data curation:** Xiaoqian Yang, Xin Wang.

**Formal analysis:** Xiaoqian Yang.

**Funding acquisition:** Xiaoqian Yang, Youcai Tuo, Yun Deng.

**Investigation:** Xiaoqian Yang, Yanjing Yang, Xin Wang.

**Methodology:** Xiaoqian Yang, Yanjing Yang.

**Project administration:** Yun Deng.

**Software:** Xiaoqian Yang, Yanjing Yang, Haoyu Wang.

**Supervision:** Youcai Tuo.

**Validation:** Xiaoqian Yang.

**Visualization:** Xiaoqian Yang, Haoyu Wang.

**Writing – original draft:** Xiaoqian Yang.

**Writing – review & editing:** Xiaoqian Yang, Youcai Tuo, Yanjing Yang, Yun Deng.

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
