## [Decision Letter · Decision Letter 0]

24 Jun 2021

PONE-D-21-16906

Study on the effect of front retaining walls on the thermal structure and outflow temperature of reservoirs.

PLOS ONE

Dear Dr. tuo,

Thank you for submitting your manuscript to PLOS ONE. After careful consideration, we feel that it has merit but does not fully meet PLOS ONE’s publication criteria as it currently stands. Therefore, we invite you to submit a revised version of the manuscript that addresses the points raised during the review process.

The paper is interesting and it could be proceeded further after major revision. 

We look forward to receiving your revised manuscript.

Kind regards,

Mohammad Mehdi Rashidi

Academic Editor

PLOS ONE

Journal Requirements:

3. Thank you for stating the following in the Competing Interests /Financial Disclosure* (delete as necessary) section:

'The authors have declared that no competing interests exist.'  

We note that one or more of the authors are employed by a commercial company: PowerChina Guiyang Survey, Design and Research Institute Co. Ltd.

4. We note that Figure 2 includes an image of a [patient / participant / in the study]. 

Additional Editor Comments (if provided):

Reviewers' comments:

Reviewer's Responses to Questions

**Comments to the Author**

1. Is the manuscript technically sound, and do the data support the conclusions?

Reviewer #1: Yes

2. Has the statistical analysis been performed appropriately and rigorously? 

Reviewer #1: Yes

3. Have the authors made all data underlying the findings in their manuscript fully available?

Reviewer #1: Yes

4. Is the manuscript presented in an intelligible fashion and written in standard English?

Reviewer #1: Yes

5. Review Comments to the Author

Reviewer #1: Title: Study on the effect of front retaining walls on the thermal structure and outflow temperature of reservoirs

MS NO. PONE-D-21-16906

In this paper, a two-dimensional hydrodynamic CE-QUALW2 model was established and calibrated to simulate the reservoir and outflow temperature with and without the front retaining wall and its impact on the thermal stability and outflow temperature are obtained. The idea seems interesting and practical; however, the following comments has to be addressed.

1. In the introduction section avoid using lumped citation and discuss the cited papers.

3. Justify why this is a 2D problem rather than a 3D one.

4. Enhance the quality of the figures.

5. Add nomenclature to the paper to enhance readability.

6. Explain why CE-QUAL-W2 model is hired.

7. Clearly justify the mismatch in the verification case.

6. PLOS authors have the option to publish the peer review history of their article (what does this mean?). If published, this will include your full peer review and any attached files.

Reviewer #1: No

---

## [Author Response · Author response to Decision Letter 0]

11 Aug 2021

We are grateful for the constructive comments and suggestions. Based on your comments, we have revised the relevant content in the manuscript.

Firstly, we improved the quality of the introduction including revising the lumped citation, summarizing the results of some academic papers, and further refining and highlighting the innovation of our paper. Secondly, we professionalized the language of the paper, and explained the reason why the dimensions and the specific models were used. Secondly, the figures and formats of the paper were modified to meet the publishing requirements of your journal.

Our point-by-point responses to your comments are provided in the Response to Reviewers.

---

## [Decision Letter · Decision Letter 1]

22 Oct 2021

PONE-D-21-16906R1Study on the effect of front retaining walls on the thermal structure and outflow temperature of reservoirs.PLOS ONE

Dear Dr. tuo,

Thank you for submitting your manuscript to PLOS ONE. After careful consideration, we feel that it has merit but does not fully meet PLOS ONE’s publication criteria as it currently stands. Therefore, we invite you to submit a revised version of the manuscript that addresses the points raised during the review process.

We look forward to receiving your revised manuscript.

Kind regards,

Mohammad Mehdi Rashidi

Academic Editor

PLOS ONE

Journal Requirements:

Reviewers' comments:

Reviewer's Responses to Questions

**Comments to the Author**

1. If the authors have adequately addressed your comments raised in a previous round of review and you feel that this manuscript is now acceptable for publication, you may indicate that here to bypass the “Comments to the Author” section, enter your conflict of interest statement in the “Confidential to Editor” section, and submit your "Accept" recommendation.

Reviewer #1: All comments have been addressed

Reviewer #2: (No Response)

2. Is the manuscript technically sound, and do the data support the conclusions?

Reviewer #1: Yes

Reviewer #2: Partly

3. Has the statistical analysis been performed appropriately and rigorously? 

Reviewer #1: Yes

Reviewer #2: N/A

4. Have the authors made all data underlying the findings in their manuscript fully available?

Reviewer #1: Yes

Reviewer #2: Yes

5. Is the manuscript presented in an intelligible fashion and written in standard English?

Reviewer #1: Yes

Reviewer #2: Yes

6. Review Comments to the Author

Reviewer #1: All comments have been addressed.

Reviewer #2: Here are some comments that author(s) need to respond:

1. Authors need to indicate following information of this research in abstract

i) The location of this research, as it is stated in section 2.1, line 85, Dongqing Reservoir located in the southwestern Guizhou Province, China. Authors need to clarify the region of proposed study clearly.

ii) The duration of collected data needs to specify, as it is provided in section 2.1, line 102-103, from January to September 2017

2. In section 3.1, results and discussion, line 172-173, authors need to provides the sources of used data in this research.

3. Please clarify that the used critical coefficients in CE-QUAL-W2 model given in Table 1, are from existing research or are fitted in this research, if yes please provide the reference.

4. Section 3.4, Engineering Applications and Prospects for this Study, the provided results in Fig 8 does not present the stated improvement in line 365-370. Authors need to explain it.

5. The quality of figures should be improved.

7. PLOS authors have the option to publish the peer review history of their article (what does this mean?). If published, this will include your full peer review and any attached files.

Reviewer #1: No

Reviewer #2: **Yes: **Touraj Khodadadi

---

## [Author Response · Author response to Decision Letter 1]

2 Nov 2021

Dear reviewer,

We are grateful for the constructive comments and suggestions. Based on your comments, we have revised the relevant content in the manuscript.

Firstly, we improved the quality of the abstract by adding to the location and duration of observation data in our research. Based on ’your opinion, we provided the source of used data, calibrated the critical coefficients given in our model and further added the cited references. Secondly, we professionalized the language of the paper, and all the figures and formats of the paper were modified to meet the publishing requirements of PLOS ONE.

Our point-by-point responses to your comments are provided below.

1): Authors need to indicate following information of this research in abstract:

i) The location of this research, as it is stated in section 2.1, line 85, Dongqing Reservoir located in the southwestern Guizhou Province, China. Authors need to clarify the region of proposed study clearly.

ii) The duration of collected data needs to specify, as it is provided in section 2.1, line 102-103, from January to September 2017.

Response to reviewer’s comment No. 1:

Thank you so much for your constructive suggestion.

Based on your suggestion, we made relative revisions on Abstract by adding the location and duration of collected data. The sentences correspondingly in line 10-14 of Manuscript was as follows:

For this purpose, taking the Dongqing Reservoir (25° 31’N, 105° 46’E) as a case study, a two-dimensional hydrodynamic CE-QUAL-W2 model was configured for the typical channel-type reservoir in the southwestern Guizhou Province, to better understand the influence of FRWs on the thermal structure and outflow temperature. The simulated data from January to September 2017 showed that FRWs can change the vertical temperature distribution during the stratification period, accelerate the upper warmer water release and thus decrease the strength of thermal stratification.

2): In section 3.1, results and discussion, line 172-173, authors need to provides the sources of used data in this research.

Response to reviewer’s comment No. 2:

Thank you so much for your constructive suggestion.

I’m sorry for the expression lacking consideration and thoroughness. Hydrological data was mainly from the operation data from the monitoring of the upstream and downstream power stations of the reservoir. The meteorological data originated from the Zhenfeng Meteorological Station, see line 172-173. Additionally, we added the sources of used data in line 168-169 of Manuscript which was as follows:

We compared the calculated results and measured data for upstream of the dam (Fig 2) and outflow temperature (Fig 3) based on hydrological and meteorological data from January to September 15, 2017, which were obtained from the monitoring of the upstream and downstream power station of the Dongqing Reservoir and the meteorological website (http://www.cma.gov.cn/2011qxfw/2011qsjgx/), respectively. 

3): Please clarify that the used critical coefficients in CE-QUAL-W2 model given in Table 1, are from existing research or are fitted in this research, if yes please provide the reference.

Response to reviewer’s comment No. 3:

Thank you for your suggestion. The critical coefficients given in Table 1 include the longitudinal eddy viscosity coefficient, longitudinal eddy diffusion coefficient, wind shielding coefficient, extinction for pure water, solar radiation absorbed in the surface layer and dynamic shading coefficient. Among them, the eddy viscosity coefficients affect the hydrodynamic conditions, which are difficult to calibrate. For the values, we refer to the case of CE-QUAL-W2 user manual [1]. Since the mountains on both sides of the bank don’t obstruct the Dongqing reservoir area, the dynamic shade coefficient and wind shade coefficient are both taken as 1 [2-4].

Therefore, we calibrate eextinctions for pure water (EXH2O) and solar radiation absorbed in the surface layer (BETA). They are related to the chromaticity and turbidity of water bodies. Studies have shown that the value of BETA ranges from 0.4 to 0.7, and EXH2O ranges from 0 to 1 [5]. The values of EXH2O are 0.3 and 0.8 respectively for calibration, and MAE and RMSE are obtained, as shown in Table 1. Similarly, choose BETA to be 0.4 for calibration, and get MAE and RMSE, see Table 2. All the calibration could be carried out under the control variables. Fig 1 and Fig 2 shows comparisons from different EXH2O and BETA in different months. 

Table 1. Calibration for EXH2O

EXH2O MAE AVE RMSE

 1 2 3 4 5 6 7 8 9 

0.8 0.15 0.25 0.34 0.39 0.55 0.50 0.86 0.71 0.43 0.46 0.69

0.5 0.14 0.25 0.31 0.36 0.49 0.46 0.86 0.69 0.44 0.44 0.63

0.3 0.14 0.25 0.27 0.33 0.54 0.52 0.83 0.70 0.43 0.45 0.64

(a) June 15, 2017 (b) August 15, 2017

Fig 1. Calibration for EXH2O in different months

Table 2. Calibration for BETA

BETA MAE AVE RMSE

 1 2 3 4 5 6 7 8 9 

0.40 0.14 0.25 0.28 0.37 0.55 0.48 1.04 0.67 0.52 0.48 0.75

0.65 0.14 0.25 0.31 0.36 0.49 0.46 0.86 0.69 0.44 0.44 0.63

(a) May 15, 2017 (b) September 15, 2017

Fig 2. Calibration for BETA in different months

Obviously, when choose 0.5 for EXH2O and 0.65 for BETA, the MAE and RMSE get smaller showing a better verification effect.

we added the sources of these parameters in line 162 of Manuscript.

Reference：

1. Cole, T.M.; Wells, S.A., 2013. CE-QUAL-W2: A Two-Dimensional, Laterally Averaged, Hydrodynamic and Water Quality Model; Version 3.71; User Manual; Department of Civil and Environmental Engineering, Portland State University: Portland, OR, USA. https://doi:10.1023/b:hydr.0000008504.61773.77

2. Xie, Q., Liu, Z., Fang, X., Chen, Y., Li, C., MacIntyre, S. 2017. Understanding the Temperature Variations and Thermal Structure of a Subtropical Deep River-Run Reservoir before and after Impoundment. Water. 9(8), 603. https://doi.org/10.11660/slfdxb.20190105

3. Yang Y, Deng Y, Tuo Y, Li J, He T, Chen M (2020) Study of the thermal regime of a reservoir on the Qinghai-Tibetan Plateau, China. PLoS ONE 15(12): e0243198. https://doi.org/10.1371/journal.pone.0243198

4. Li, J., Liu, S., Sun, D., Yang, J. 2015. Discussion on the application of CE-QUAL-W2 water temperature model for long and narrow valley reservoirs [C]. 25th Chinese Society For Environmental Sciences Conference. 1652-1658.

5. Deng Y. 2003. Study on the Water Temperature Prediction Model for the Huge and Deep Reservoir. PhD dissertation, Sichuan University. https://kns.cnki.net/KCMS/detail/detail.aspx?dbname=CDFD9908&filename=2004030814.nh

4): Section 3.4, Engineering Applications and Prospects for this Study, the provided results in Fig 8 does not present the stated improvement in line 365-370. Authors need to explain it.

Response to reviewer’s comment No. 4:

Thank you so much for your suggestion. We have revised Figure 8 by adding to the whole withdrawal elevations from March to September. It can be seen that the minimum elevation is 472.5m in June when there is a front retaining wall before the dam. Usually, the withdrawal layer referring to the layer with the maximum flow rate is recognized as the mean plug layer (Part 3.3.2, line 283). The water level of the Dongqing Reservoir and the elevation of the FRW is mentioned in line 90 and line 120 of Manuscript considerably.

Moreover, we reorganized our texts in line 326-329:

For example, the water level of the Dongqing reservoir ranges from 484m to 490m. The lowest elevation of the mean plug flow is 472.5 m (June, Fig 8) near the upper thermocline, causing that the upper high-temperature water body can be continuously obtained.

Fig 8. Elevations of the withdrawal layer in different months and different working conditions and the local flow field distribution in front of the dam in April, July and September

5): The quality of figures should be improved.

Response to reviewer’s comment No. 5:

Thank you for your kindly reminder. We complemented all the figures with corresponding parts including enlarging the font, improving the clarity, readability and beauty of the figures. We employed 300dpi of all the figures as output parameter with Adobe Illustrator CC software. And then we upload our figure files to the Preflight Analysis and Conversion Engine (PACE) digital diagnostic tool to ensure that figures meet PLOS ONE requirements.

Once again, thank you for your valuable suggestions and comments that enable us to substantially improve this paper. We hope the corrections will be accepted.

Best wishes,

Corresponding Author: Associate Prof. Youcai, Tuo

2021-11-2

---

## [Decision Letter · Decision Letter 2]

17 Nov 2021

Study on the effect of front retaining walls on the thermal structure and outflow temperature of reservoirs.

PONE-D-21-16906R2

Dear Dr. tuo,

We’re pleased to inform you that your manuscript has been judged scientifically suitable for publication and will be formally accepted for publication once it meets all outstanding technical requirements.

Kind regards,

Mohammad Mehdi Rashidi

Academic Editor

PLOS ONE

Additional Editor Comments (optional):

Reviewers' comments:

Reviewer's Responses to Questions

**Comments to the Author**

1. If the authors have adequately addressed your comments raised in a previous round of review and you feel that this manuscript is now acceptable for publication, you may indicate that here to bypass the “Comments to the Author” section, enter your conflict of interest statement in the “Confidential to Editor” section, and submit your "Accept" recommendation.

Reviewer #2: All comments have been addressed

2. Is the manuscript technically sound, and do the data support the conclusions?

Reviewer #2: Yes

3. Has the statistical analysis been performed appropriately and rigorously? 

Reviewer #2: Yes

4. Have the authors made all data underlying the findings in their manuscript fully available?

Reviewer #2: Yes

5. Is the manuscript presented in an intelligible fashion and written in standard English?

Reviewer #2: Yes

6. Review Comments to the Author

Reviewer #2: (No Response)

7. PLOS authors have the option to publish the peer review history of their article (what does this mean?). If published, this will include your full peer review and any attached files.

Reviewer #2: **Yes: **Touraj khodadadi

---

## [Editor Report · Acceptance letter]

1 Dec 2021

PONE-D-21-16906R2 

Study on the effect of front retaining walls on the thermal structure and outflow temperature of reservoirs 

Dear Dr. Tuo:

I'm pleased to inform you that your manuscript has been deemed suitable for publication in PLOS ONE. Congratulations! Your manuscript is now with our production department. 

Kind regards, 

on behalf of

Professor Mohammad Mehdi Rashidi 

Academic Editor

PLOS ONE